# Clonal dynamics following *p53* loss of heterozygosity in *Kras*-driven cancers

Mandar Deepak Muzumdar[1,2,3], Kimberly Judith Dorans[1], Katherine Minjee Chung[1], Rebecca Robbins[1], Tuomas Tammela[1], Vasilena Gocheva[1], Carman Man-Chung Li[1,4] & Tyler Jacks[1,4,5]

Although it has become increasingly clear that cancers display extensive cellular heterogeneity, the spatial growth dynamics of genetically distinct clones within developing solid tumours remain poorly understood. Here we leverage mosaic analysis with double markers (MADM) to trace subclonal populations retaining or lacking p53 within oncogenic *Kras*-initiated lung and pancreatic tumours. In both models, p53 constrains progression to advanced adenocarcinomas. Comparison of lineage-related *p53* knockout and wild-type clones reveals a minor role of p53 in suppressing cell expansion in lung adenomas. In contrast, p53 loss promotes both the initiation and expansion of low-grade pancreatic intraepithelial neoplasia (PanINs), likely through differential expression of the p53 regulator p19ARF. Strikingly, lineage-related cells are often dispersed in lung adenomas and PanINs, contrasting with more contiguous growth of advanced subclones. Together, these results support cancer type-specific suppressive roles of p53 in early tumour progression and offer insights into clonal growth patterns during tumour development.

[1] David H. Koch Institute for Integrative Cancer Research, Massachusetts Institute of Technology, 77 Massachusetts Avenue 75-453, Cambridge, Massachusetts 02139, USA. [2] Dana-Farber Cancer Institute, Boston, Massachusetts 02215, USA. [3] Harvard Medical School, Boston, Massachusetts 02115, USA. [4] Department of Biology, Massachusetts Institute of Technology, Cambridge, Massachusetts 02139, USA. [5] Howard Hughes Medical Institute, Massachusetts Institute of Technology, Cambridge, Massachusetts 02139, USA. Correspondence and requests for materials should be addressed to T.J. (email: tjacks@mit.edu).

Cancer cells within developing tumours exhibit significant genetic and phenotypic heterogeneity mediating tumour growth, metastasis and therapy resistance[1–3]. This intratumoral heterogeneity is thought to arise from the sequential accumulation of genetic or epigenetic changes that favour the growth of distinct subclonal populations. Indeed, construction of genetic hierarchies from genomic sequencing data reveals the presence of subclonal populations within individual tumours that propagate throughout progression from early to advanced primary tumours and metastases[4–7]. Studies in transplant models have underscored the functional importance of specific genetic variants in modulating growth dynamics of different subclones within tumours[8,9]. Unfortunately, similar analyses in physiologically relevant, autochthonous cancer models during tumour progression are lacking[10] due to technical challenges in inducing sequential mutations in subclonal populations and unambiguously tracing them at single-cell resolution.

We have previously developed autochthonous models of lung and pancreatic cancer by simultaneous Cre recombinase-mediated activation of oncogenic $Kras$ ($Kras^{G12D}$) and biallelic inactivation of $p53$ in cells residing in the tissues of origin[11–13]. These models faithfully recapitulate certain prevalent genetic alterations, histologic tumour progression, metastatic behaviour and treatment response of the human diseases. By comparing $LSL\text{-}Kras^{G12D}/Kras^{WT}$; $p53^{WT/WT}$ and $LSL\text{-}Kras^{G12D}/Kras^{WT}$; $p53^{flox/flox}$ mice infected with inhaled adenovirus carrying Cre recombinase, our laboratory revealed a role of p53 in limiting tumour progression from low-grade lung adenomas to advanced adenocarcinomas[11]. Furthermore, reactivation of p53 in advanced lung tumours led to selective loss of adenocarcinoma cells[14,15], consistent with a specific role of $p53$ mutation in regulating late-stage lung tumour progression. Finally, exome-sequencing analyses of murine lung adenocarcinomas derived from $LSL\text{-}Kras^{G12D}/Kras^{WT}$; $p53^{flox/flox}$ mice revealed no recurrent mutations beyond $Kras$ and $p53$ (ref. 16), suggesting that $p53$ loss is the main genetic driver of tumour progression in this model.

Previous studies have also suggested that p53 principally plays a role late in pancreatic tumorigenesis. Similar to what is seen in human lung tumours[17], $p53$ mutations are primarily observed in more advanced human pancreatic lesions, including pancreatic ductal adenocarcinoma (PDAC) or precursor PanINs of high-grade histology[18,19]. Moreover, $p53$ mutation shortens the latency and increases the frequency of PDAC formation in mouse pancreatic tumour models in which $p53$ is simultaneously mutated at the time of oncogenic $Kras$ activation[13,20].

In this study, we adapt these models to permit sequential and sporadic $p53$ loss of heterozygosity (LOH) following oncogenic $Kras$-mediated tumour initiation. We more faithfully model clonal evolution during tumorigenesis and perform high-resolution tracing of subclones lacking or retaining p53 during tumour progression. We demonstrate that sporadic $p53$ loss promotes progression to advanced lung and pancreatic tumours. Moreover, we confirm that p53 primarily plays a role late in lung tumorigenesis. In contrast, we determine that p53 suppresses both the initiation and expansion of early pancreatic tumours, which correlates with expression of the p53 regulator p19ARF. Finally, we show surprisingly significant intratumoral cell dispersion of subclones in early lung and pancreatic tumours.

## Results

**Induction of $p53$ LOH using MADM in mice.** To generate sporadic $p53$ LOH in $Kras$-initiated tumors, we took advantage of mosaic analysis with double markers (MADM), which permits simultaneous fluorescence cell labelling and mutagenesis through a single Cre-mediated inter-chromosomal recombination event in

mice[21]. MADM has been used to study the consequence of tumour suppressor gene LOH on tissue development and cancer initiation at single-cell resolution[22,23]. We crossed $LSL\text{-}Kras^{G12D}$ mice with $MADM11\text{-}GT,p53^{WT}/MADM11\text{-}TG,p53^{KO}$ mice to generate $LSL\text{-}Kras^{G12D}/Kras^{WT}$; $MADM11\text{-}GT,p53^{WT}/MADM11\text{-}TG\text{-}p53^{KO}$ mice ($K\text{-}MADM\text{-}p53$) (Methods). On Cre expression, oncogenic $Kras$ is efficiently induced via intra-chromosomal Cre-mediated recombination permitting tumour initiation (Fig. 1a). Sporadic $p53$ LOH occurs by subsequent stochastic and inefficient Cre-mediated inter-chromosomal recombination between homologous chromosomes. Mitotic recombination and X segregation (G2-X) of the MADM cassettes is predicted to result in the generation of two genotypically and phenotypically distinct daughter cells from a colourless $p53^{KO/WT}$ parent cell: GFP + /tdTomato − (green) $p53^{KO/KO}$ and GFP − /tdTomato + (red) $p53^{WT/WT}$ (Fig. 1a,b and Supplementary Fig. 1). In contrast, G2-Z, G0 or G1 recombination results in the generation of GFP + /tdTomato + (yellow) and GFP − /tdTomato − (colourless) $p53^{KO/WT}$ cells (Fig. 1b,c and Supplementary Fig. 1). As the fluorescent markers are genetically encoded, the MADM system affords tracing of lineage-related green $p53^{KO/KO}$ and red $p53^{WT/WT}$ subclones, allowing for the induction and monitoring of intratumoral heterogeneity in autochthonous tumours.

**Sporadic $p53$ LOH promotes progression to lung adenocarcinoma.** To determine whether sporadic and sequential (following $Kras$ mutation) $p53$ LOH promotes lung tumour progression, we administered lentiviral Cre via the trachea to adult $K\text{-}MADM\text{-}p53$ mice to induce stable Cre expression in lung epithelial cells[24]. Infected mice exhibited multiple small lung tumours containing fluorescently labelled cells, although green ($p53^{KO/KO}$) cells did not predominate at early time points (Fig. 2a). Mice analysed at later time points, however, displayed an increase in the overall size of tumours and the development of larger and more numerous green tumours. Histologic analysis of these large tumours revealed high-grade lesions (adenocarcinomas) consisting of densely packed green cells (Fig. 2b). We also observed mixed-grade tumours (mixed adenoma–adenocarcinomas) in which the adenocarcinoma component was entirely green (Fig. 2c). These data suggest that the sequential loss of $p53$ is a driver of tumour progression to adenocarcinoma in oncogenic $Kras$-initiated lung tumours.

We confirmed these findings using an alternative, less efficient MADM model in which $Kras$-initiated lung tumours spontaneously arise through a Cre-independent stochastic recombination event ($Kras^{LA2}$ model)[25] and MADM-labelled clones are thereafter generated by tamoxifen-induced Cre activation (Cre$^{ERT2}$) (Fig. 3a). This method ensures sequential mutation of $p53$ following tumour initiation by oncogenic $Kras$. From eight $Kras^{LA2},Rosa26\text{-}Cre^{ERT2}/Kras^{WT}$; $MADM\text{-}p53$ mice dissected following the development of tumour-related morbidity, we observed two fluorescently labelled tumours on whole mount analysis (Fig. 3b). These tumours were green ($p53^{KO/KO}$) and displayed histologic features of adenocarcinoma (Fig. 3c). In addition, a small number of low-grade adenomas harboured rare yellow $p53^{KO/WT}$ cells (Fig. 3d), supporting the inefficient nature of MADM recombination in this model and the clonality of the green tumour cells. No fluorescently labelled tumours were observed on whole mount analysis of lungs from ten $Kras^{LA2},Rosa26\text{-}Cre^{ERT2}/Kras^{WT}$; $MADM$ mice (lacking $p53$ mutation). Together, these data are consistent with a role of p53 in constraining lung tumour progression to adenocarcinoma.

**$p53$ loss does not greatly impact early lung tumorigenesis.** To further evaluate whether $p53$ also suppresses cell expansion in

early lung tumours, we classified lung adenomas based on their green or red cell predominance in tissue sections of *K-MADM-p53* mice at 10–16 weeks post infection (p.i.) (Fig. 4a). As green and red cells are produced at 1:1 stoichiometry following a single G2-X recombination event (Fig. 1a), green cells should out-number red cells in these tumours (green-dominant) if *p53* loss promotes cell expansion during early tumorigenesis. In contrast, a plurality of tumours (51 of 132) showed no colour dominance on qualitative analysis of random cross-sections of adenomas. Given the possibility for stochastic differences in individual daughter cell expansion following G2-X recombination, we did observe tumours (69 of 132) that showed green or red cell predominance. However, the proportions of green-dominant and red-dominant tumours were not statistically different, suggesting that green $p53^{KO/KO}$ cells did not have a selective growth

advantage at this stage (Fig. 4b). To more rigorously characterize the ratio of green to red cells (green-to-red ratio) within individual tumours, we quantified labelled cells across serial sections through entire lung adenomas derived from mice dissected 10 or 16 weeks p.i. Again, we observed no difference in the intratumoral proportions of $p53^{KO/KO}$ and $p53^{WT/WT}$ cells at 10 weeks p.i. and only a small difference at 16 weeks p.i. (Fig. 4c,d). These results indicate that *p53* loss does not significantly affect tumour cell expansion in lung adenomas.

Given the stochastic nature of mitotic recombination events leading to MADM labelling, we were unable to definitively determine the timing of *p53* loss in *K-MADM-p53* mice. To circumvent this limitation, we used the sum of all red and green single-labelled cells in a tumour as a surrogate for timing of G2-X recombination with the assumption that increased overall cell

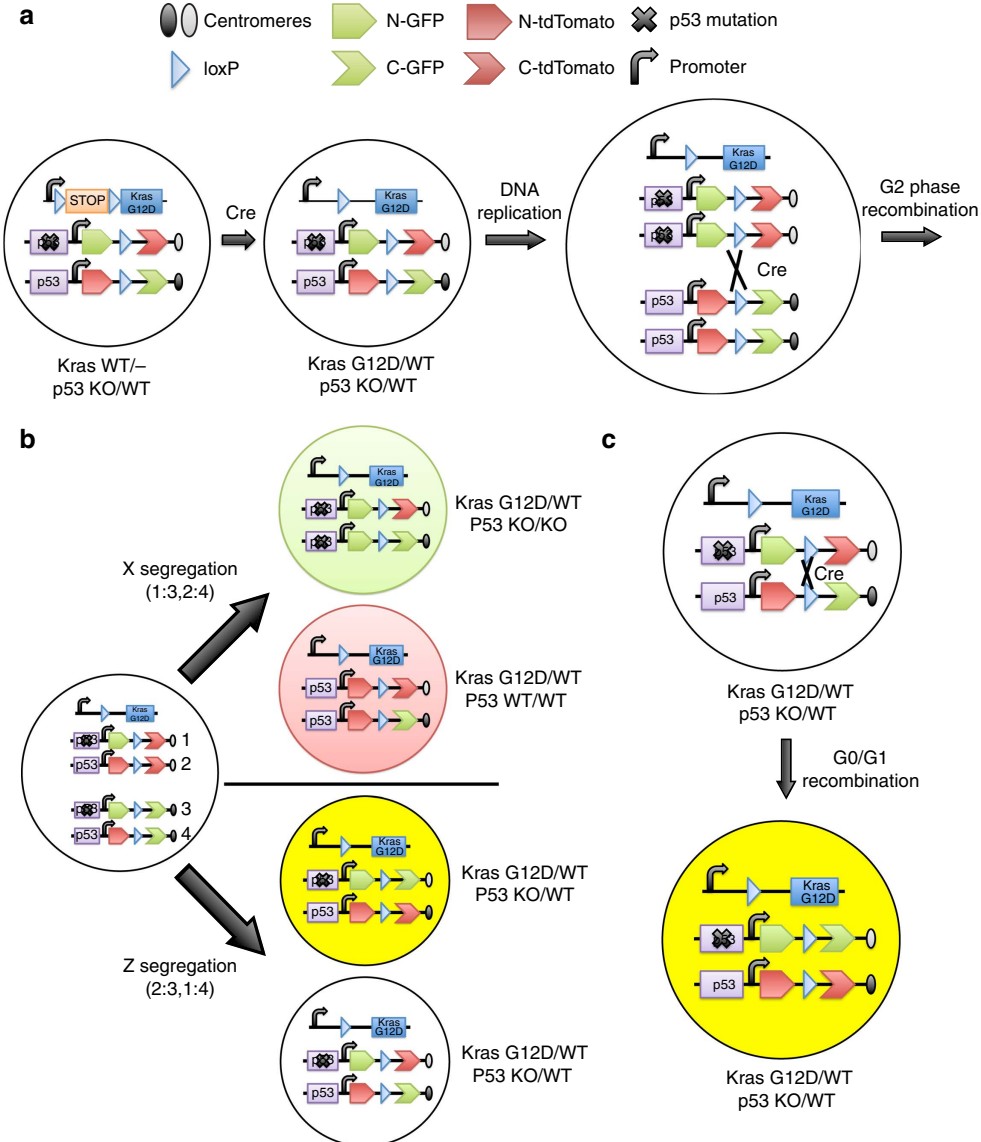

**Figure 1 | Schematic of MADM system.** (**a**) Schematic of MADM-mediated LOH of *p53*. Efficient Cre-mediated intra-chromosomal recombination deletes the transcriptional/translational STOP cassette inducing oncogenic *Kras* activation. Less efficient Cre-mediated inter-chromosomal recombination following DNA replication (during G2 phase) leads to reconstitution of GFP and tdTomato on separate chromosomes before cell division. This diagram was adapted with permission from the original MADM schematic[21]. (**b**) X segregation of chromosomes following mitotic recombination (G2-X) results in genetically distinct daughter cells: $p53^{KO/KO}$ (green, GFP + /tdTomato − ) and $p53^{WT/WT}$ (red, GFP − /tdTomato + ) cells. Z-segregation (G2-Z) leads to the generation of yellow (GFP + /tdTomato + ) and colourless (GFP − /TdTomato − ) $p53^{KO/WT}$ cells. (**c**) Cre-mediated inter-chromosomal recombination during G0 or G1 phase results in the production of yellow $p53^{KO/WT}$ from colourless $p53^{KO/WT}$ cells. The MADM system affords faithful correlation between the expression of a specific genetically encoded fluorescence marker and genotype.

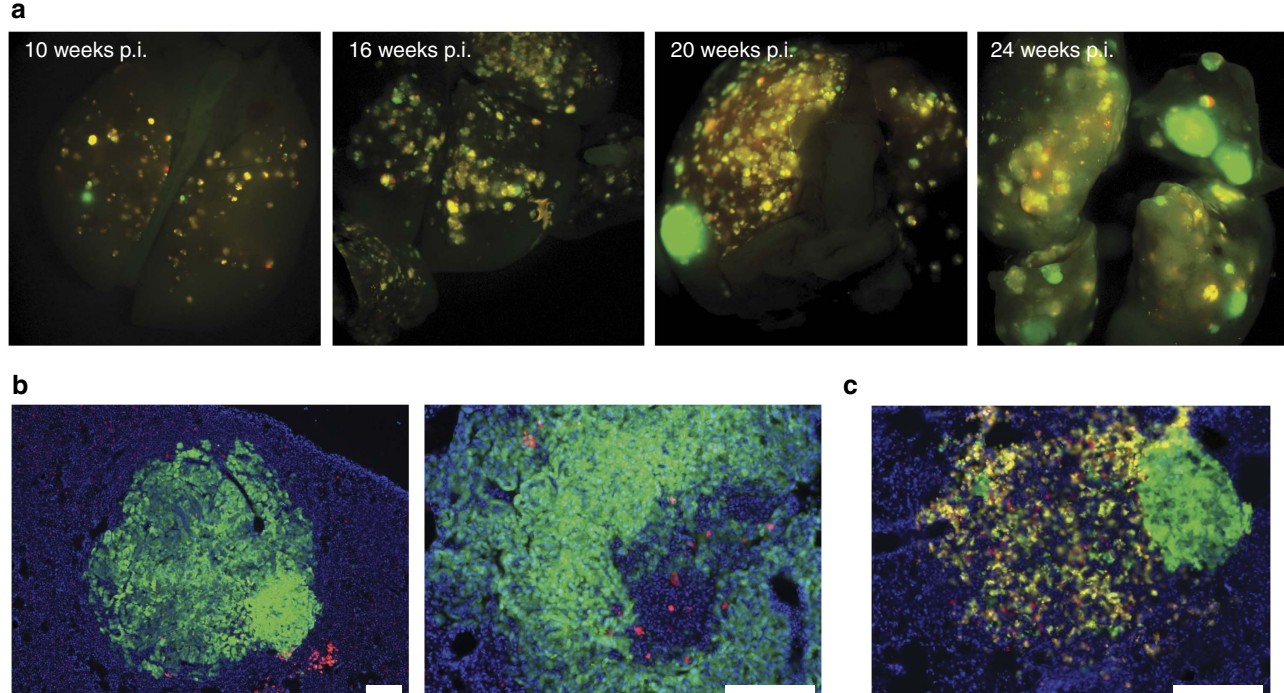

**Figure 2 | p53 constrains lung tumour progression in the *LSL-Kras*^G12D-MADM model.** (**a**) Whole-mount images of *K-MADM-p53* mice at various time points following lentiviral Cre administration displaying *p53*^KO/KO (green, GFP+/tdTomato-), *p53*^WT/WT (red, GFP-/tdTomato+) and *p53*^KO/WT (yellow, GFP+/tdTomato+) cells within lung tumours. p.i., post infection. (**b**) Lung adenocarcinomas consisting mostly of green cells. (**c**) Mixed-grade tumour consisting of adenoma (left) and adenocarcinoma cells (right). The adenocarcinoma component consists of all green cells. Blue, DAPI-stained nuclei. Scale bars, 200 μm (all).

labelling indicates earlier time points of *p53* LOH. If the duration of *p53* loss altered cell expansion, we would expect a positive correlation between the total number of single-labelled cells and the green-to-red ratio. Interestingly, there was no association between these two parameters in lung adenomas (Fig. 4e). Together, these data confirm earlier work[11,14,15] demonstrating that *p53* loss does not have a significant impact on early lung tumorigenesis.

**p53 LOH drives tumour progression to PDAC.** To evaluate the effect of *p53* LOH on pancreatic tumour progression using MADM, we crossed *K-MADM-p53* mice with *Pdx1-Cre* mice to direct Cre expression to the developing pancreas[12]. *Pdx1-Cre-MADM-p53* mice (lacking *LSL-Kras*^G12D) exhibited green, red and yellow acinar, ductal and islet cells but no overt cellular phenotypes due to *p53* loss (Fig. 5a and Supplementary Fig. 2). In contrast, *Pdx1-Cre-K-MADM-p53* mice developed the full spectrum of pancreatic tumour progression from low-grade (Fig. 5b) and high-grade PanINs (Fig. 5c) to advanced PDAC (Fig. 5d) and occasionally distant metastases (Fig. 5e,f). Interestingly, *Pdx1-Cre-K-MADM-p53* mice exhibited a median survival of ~11 weeks, falling in between that observed in *Pdx1-Cre; LSL-Kras*^G12D/*Kras*^WT (KC) mice harbouring homozygous *p53* mutation (~6 weeks) and heterozygous *p53* mutation (~16 weeks) (Fig. 5g), supporting *p53* LOH as an important driver of tumour progression in this model. Consistent with p53 constraining progression to advanced disease, high-grade PanINs and PDACs were predominantly or completely green at an intermediate time point (6 weeks) (Fig. 6a,b). We confirmed this green predominance of advanced lesions in intact pancreata using CLARITY tissue clearing[26] (Fig. 6c,d).

**p53 suppresses PanIN initiation and expansion.** To determine the consequence of *p53* loss on tumour initiation in pancreatic

cancer, we took advantage of the fact that G2-X recombination and labelling could occur during pancreatic development (as *Pdx1-Cre* is expressed as early as E8.5 (ref. 12)) before transformation, resulting in PanINs comprising cells of uniform colour. We first confirmed that there was no difference in the proportions of green and red normal duct cells, the putative cell-of-origin for PanINs[19]. Next, we quantified the number of low-grade PanINs harbouring all green or all red cells in 6-week-old *Pdx1-Cre-K-MADM-p53* mice (Fig. 7a). Interestingly, we observed a greater frequency of all-green PanINs (Fig. 7b), suggesting that *p53* loss promoted pancreatic tumour initiation by oncogenic *Kras*. We also evaluated the role of p53 on cell expansion in low-grade tumours by analysing the proportion of incompletely labelled low-grade PanINs (G2-X recombination occurring after tumour initiation) containing predominantly green or red cells (Fig. 7c). Unlike our observations in early lung tumours, we found increased numbers of green-dominant compared with red-dominant low-grade PanINs (Fig. 7d), consistent with enhanced cell expansion following *p53* loss in early pancreatic tumours. Overall, these findings suggest a potential tumour suppressive role of p53 throughout oncogenic *Kras*-mediated pancreatic tumorigenesis, contrasting with mainly late functions during lung tumour progression.

To explore the mechanism behind p53-mediated suppression of cell expansion, we assessed proliferation (pulse EdU incorporation) and apoptosis (cleaved caspase-3) by immunostaining tissue sections of pancreatic and lung tumours. The percentage of *p53*^KO/KO cells exhibiting EdU incorporation was significantly increased compared with *p53*^WT/WT cells in low-grade PanINs (Supplementary Fig. 3a). In contrast, apoptosis was rare, largely limited to cells detached into the lumen and not related to cells of a particular *p53* genotype (Supplementary Fig. 3b). As low-grade lung tumours displayed low levels of overall proliferation, few *p53*^WT/WT and *p53*^KO/KO cells were co-labelled with EdU with no

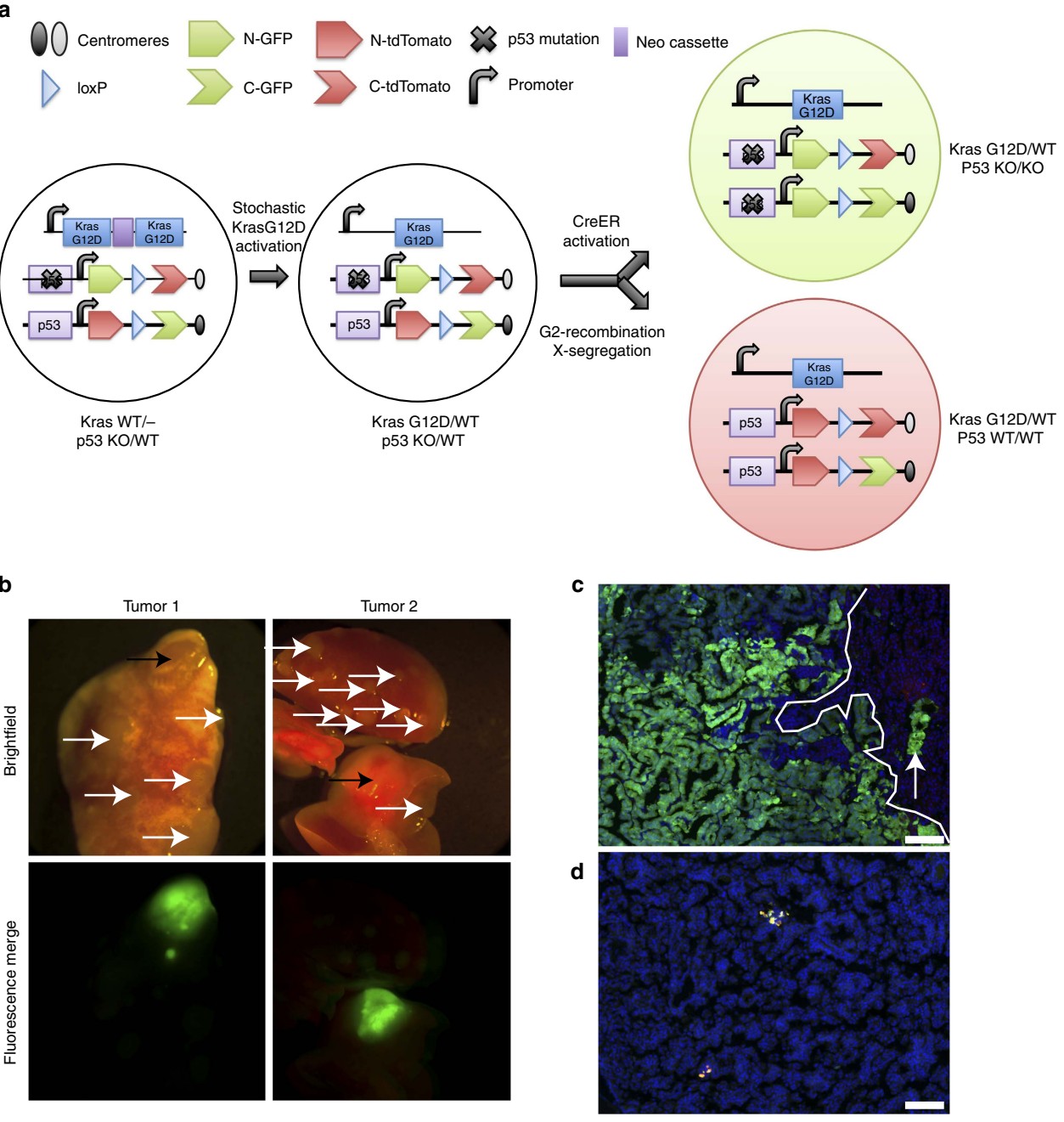

**Figure 3 | p53 constrains lung tumour progression in the *Kras^LA2*-MADM model.** (**a**) Schematic of MADM-mediated LOH of *p53* in *Kras^LA2*, *Rosa26-Cre^ERT2/Kras^WT; MADM-p53* mice. Stochastic recombination results in removal of one of two duplicate copies of mutant *Kras* exon1 (*Kras^G12D*) and an intervening neo cassette permitting expression of mutant *Kras* expression and tumour initiation[25]. G2-X MADM recombination, resulting in *p53^KO/KO* (green, GFP +/tdTomato −) and *p53^WT/WT* (red, GFP −/tdTomato +) cells, is initiated through tamoxifen activation of Cre^ERT2, permitting localization of Cre to the nucleus. This diagram was adapted with permission from the original MADM schematic[21]. (**b**) Two green tumours (black arrows) were observed on whole-mount analysis of lungs from *Kras^LA2, Rosa26-Cre^ERT2/Kras^WT; MADM-p53* mice (*n* = 8), whereas none were observed in *Kras^LA2,Rosa26-Cre^ERT2/ Kras^WT; MADM* mice (not harbouring *p53* mutation, *n* = 10). White arrows denote tumors without fluorescence labelling. We did not detect any red or yellow tumours in either cohort of mice by whole-mount analysis. Merged fluorescence images of green and red filters are shown. (**c**) Histologic section of a tumour in **b** showed green adenocarcinoma cells adjacent to colourless adenoma cells (predominantly to the right of the line). Some green adenocarcinoma cells (arrow) are intercalating in the adenoma area. Blue, DAPI-stained nuclei. Scale bar, 100 μm. (**d**) *Kras^LA2,Rosa26-Cre^ERT2/Kras^WT; MADM-p53* adenoma harbouring rare yellow cells. Blue, DAPI-stained nuclei. Scale bar, 100 μm.

obvious difference in the percentages of labelled cells (Supplementary Fig. 3c). High-grade *p5^KO/KO* tumours showed much greater EdU incorporation (Supplementary Fig. 3d,e), whereas apoptosis was not observed in lung tumours (Supplementary Fig. 3f). Together, these data suggest that *p53* loss promotes cell cycle progression in early pancreatic tumours.

**Differential expression of p19ARF-p53 during tumorigenesis.** We hypothesized that differences in the timing of induction or stabilization of p53 protein expression may account for the functional differences observed between the tumour types. As wild-type p53 is difficult to detect by immunohistochemistry (IHC) on tissue sections with currently available antibodies,

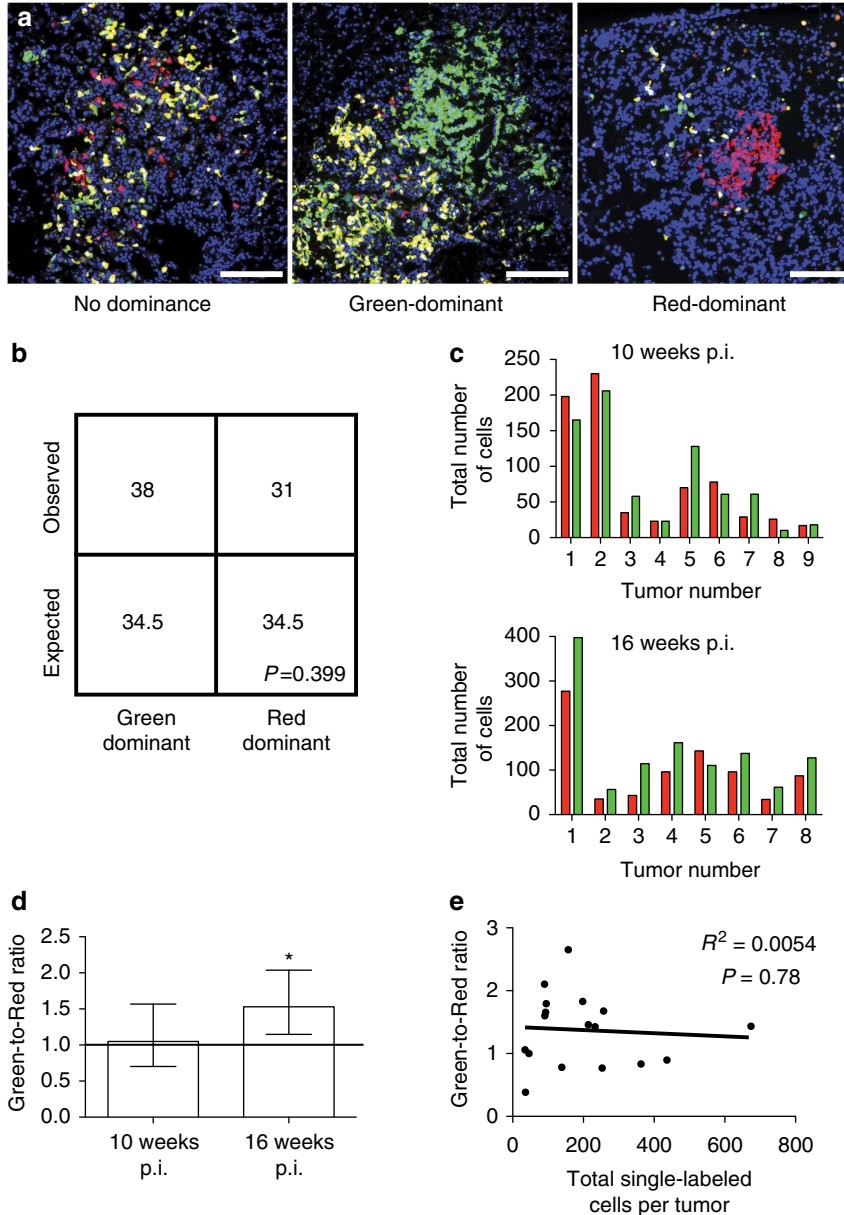

**Figure 4 | *p53* loss does not significantly have an impact on early lung tumorigenesis.** (**a**) Lung adenomas from *K-MADM-p53* mice 16 weeks p.i. showing varying degrees of $p53^{KO/KO}$ (green, GFP +/tdTomato − ), $p53^{WT/WT}$ (red, GFP −/tdTomato + ) and $p53^{KO/WT}$ (yellow, GFP +/tdTomato + ) cell labelling. Representative images of non-dominant, green-dominant and red-dominant tumours are shown. Blue, DAPI-stained nuclei. Scale bars, 200 μm (all). (**b**) Absolute quantification of observed green-dominant and red-dominant lung adenomas in *K-MADM-p53* mice (10–16 weeks p.i., $n = 5$ mice total). Expected numbers are based on 1:1 stoichiometric ratio of green and red cell generation, and stochastic growth thereafter. No statistical difference was observed ($P > 0.05$, $\chi^2$-test). A plurality of tumours (51 of 132) did not exhibit colour dominance. (**c**) Absolute quantification of green and red cells across individual tumours derived from *K-MADM-p53* mice evaluated at 10 weeks p.i. ($n = 9$) and 16 weeks p.i. ($n = 8$). (**d**) Geometric means ( ± 95% confidence intervals) of green-to-red cell ratio in lung tumours (based on data from **c**, $n = 9$ at 10 weeks p.i. and $n = 8$ at 16 weeks p.i.). Line represents equal green and red cell numbers (ratio = 1). The green-to-red cell ratio is mildly but significantly increased in tumours from 16-week-old mice (* denotes 95% confidence interval does not cross unity). (**e**) No statistically significant correlation between green-to-red cell ratio and total single-labelled (green plus red) cells per tumour was observed ($P > 0.05$, linear regression).

we took advantage of oncogenic *Kras*-driven lung and pancreatic cancer models harbouring a $p53^{R172H}$ mutant allele (*LSL-Kras$^{G12D}$*; *LSL-p53$^{R172H}$*), which demonstrate similar histologic progression to the MADM models[11,20]. In these mice, mutant *p53* is stabilized, due in part to loss of feedback inhibition, and serves as a marker of endogenous p53 expression[27]. Consistent with our hypothesis, we observed p53 protein expression in pancreatic but not lung cells during all stages of tumorigenesis from acinar-to-ductal metaplasia and low-grade PanINs to advanced disease (Fig. 8).

Previous work from our laboratory has suggested that tissue-specific expression of p19ARF, a positive upstream regulator of p53, could alter the response to oncogenic *Kras* in tumour initiation[28]. Using *LSL-Kras$^{G12D}$*; *LSL-p53$^{R172H}$* mice, we observed expression of p19ARF in early- and late-stage pancreatic lesions in similar pattern to p53 expression (Fig. 9a). In contrast, lung adenocarcinomas, but not adenomas, expressed p19ARF (Fig. 9b). As *p53* mutant cells may induce p19ARF by loss of negative feedback[28], we verified that p19ARF expression

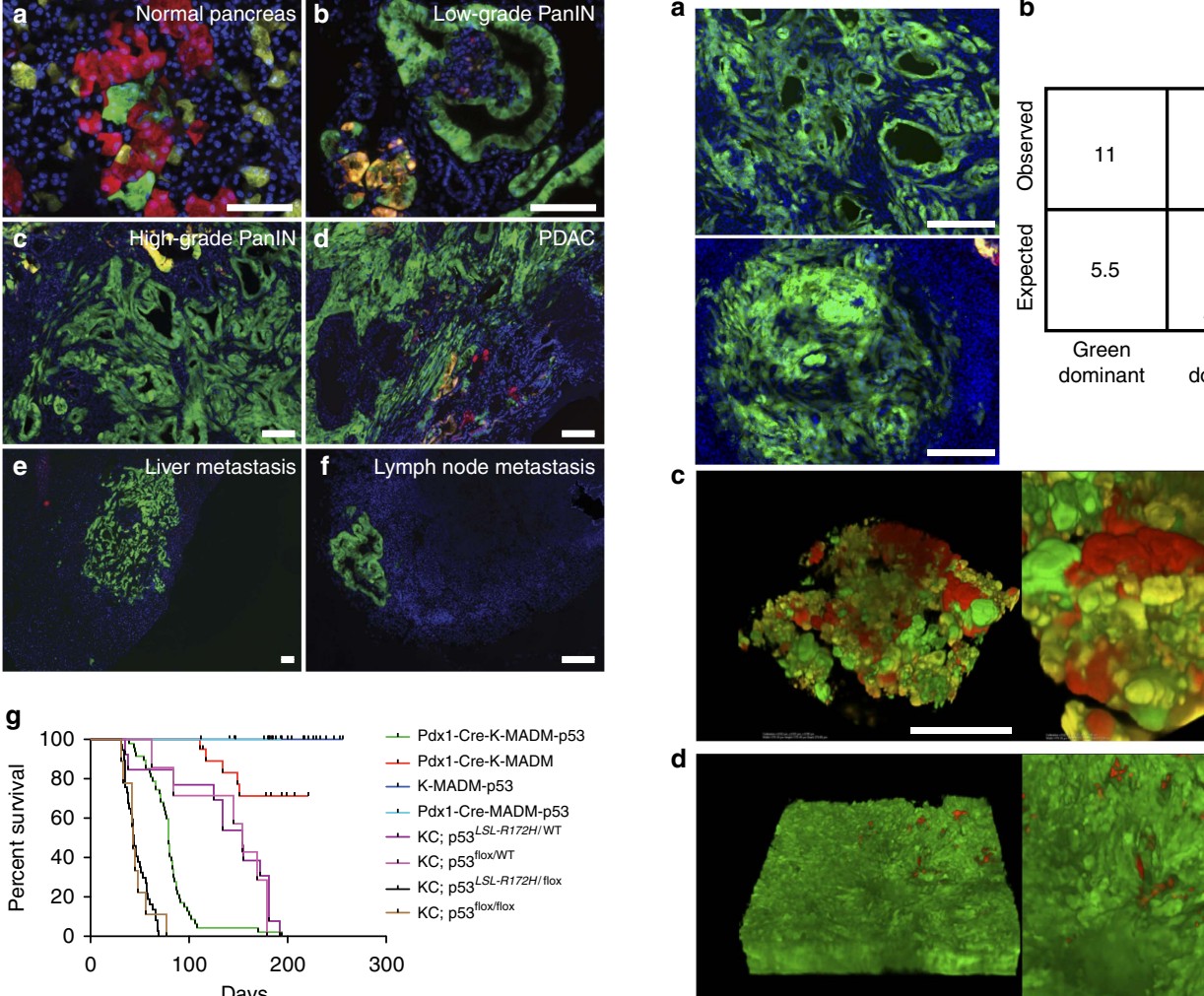

**Figure 5 | p53 LOH promotes pancreatic tumour progression.**
(**a**) Pancreas from a 5-week-old *Pdx1-Cre-MADM-p53* mouse contains $p53^{KO/KO}$ (green, GFP +/tdTomato −), $p53^{WT/WT}$ (red, GFP −/ tdTomato +) and $p53^{KO/WT}$ (yellow, GFP +/tdTomato +) acinar cells with normal appearance. (**b**) Low-grade pancreatic intraepithelial neoplasm (PanIN) in an 8-week-old *Pdx1-Cre-K-MADM-p53* mouse. (**c**) High-grade PanIN in a 6-week-old *Pdx1-Cre-K-MADM-p53* mouse. (**d**) PDAC in an 11-week-old *Pdx1-Cre-K-MADM-p53* mouse. (**e**) Liver metastasis in a 10-week-old *Pdx1-Cre-K-MADM-p53* mouse harbouring primary PDAC tumours in the pancreas. (**f**) Lymph node metastasis in a 10-week-old *Pdx1-Cre-K-MADM-p53* mouse harbouring primary PDAC tumours in the pancreas. (**g**) Kaplan–Meier survival analysis of *Kras* and *p53* mutant mouse models of PDAC. *Pdx1-Cre-K-MADM-p53* mice (*n* = 47, median survival 79 days) harbour intermediate median survival between *Pdx1-Cre; LSL-Kras^{G12D}/Kras^{WT}* (*KC*) homozygous *p53* mutant (*KC; p53^{flox/flox}* (*n* = 9, median survival 43 days) and *KC; p53^{LSL-R172H/flox}* (*n* = 37, median survival 44 days)) and heterozygous *p53* mutant (*KC; p53^{flox/WT}* (*n* = 7, median survival 154 days) and *KC; p53^{LSL-R172H/WT}* (*n* = 13, median survival 154 days)) models. *Pdx1-Cre-K-MADM* mice (with wild-type p53, *n* = 20) have prolonged survival similar to that observed with *KC* mice lacking *p53* mutation[12]. Control *Pdx1-Cre-MADM-p53* (lacking *LSL-Kras^{G12D}*, *n* = 21) and *K-MADM-p53* (lacking *Pdx1-Cre*, *n* = 16) mice do not initiate pancreatic tumorigenesis. Blue, DAPI-stained nuclei. Scale bars, 100 μm (all). (**a**,**b**) were reproduced with permission from MIT.

**Figure 6 | p53 constrains pancreatic tumour progression in *Pdx1-Cre-K-MADM-p53* mice.** (**a**) Representative images of high-grade PanIN and PDAC from 6-week-old *Pdx1-Cre-K-MADM-p53* mice show uniform green (GFP +/tdTomato −, $p53^{KO/KO}$) labelling in these advanced lesions. Blue, DAPI-stained nuclei. Scale bars, 100 μm. (**b**) Absolute quantification of high-grade PanINs/PDAC in 6-week-old *Pdx1-Cre-K-MADM-p53* mice (*n* = 5 mice total). Expected numbers are based on 1:1 stoichiometric ratio of green and red cell generation, and were not observed (*P* < 0.001, $\chi^2$-test). (**c**) CLARITY image of pancreas from a 13-week-old *Pdx1-Cre-K-MADM-p53* mouse showing grossly diseased pancreas (PanINs interspersed with normal pancreatic tissue) without overt PDAC. Right image shows zoomed in area from left image. Scale bars, 1 mm (left image) and 100 μm (right image). (**d**) CLARITY image of PDAC from a 13-week-old *Pdx1-Cre-K-MADM-p53* mouse. Right image shows zoomed in area from left image. Scale bars, 1 mm (left image) and 100 μm (right image).

was observed in early pancreatic tumours even in the context of wild-type *p53* (Fig. 9c). These data suggest that the p19ARF-p53 axis may play a role in suppressing early pancreatic

tumorigenesis. This observation is compatible with the decreased capacity of oncogenic *Kras* to initiate pancreatic tumours compared with lung tumours in mice[29]. When we examined the lungs and pancreata of *Kras^{LA2}* mice, which undergo stochastic somatic activation of oncogenic *Kras* throughout the mouse[25], 48/48 (100%) mice harboured lung tumours, but only 1/48 (2%) had PanIN lesions.

**Extratumoral invasion of PanIN cells**. In addition to pursuing quantitative analyses of tumour cell expansion, we also used the MADM models to monitor the spatial relationships between

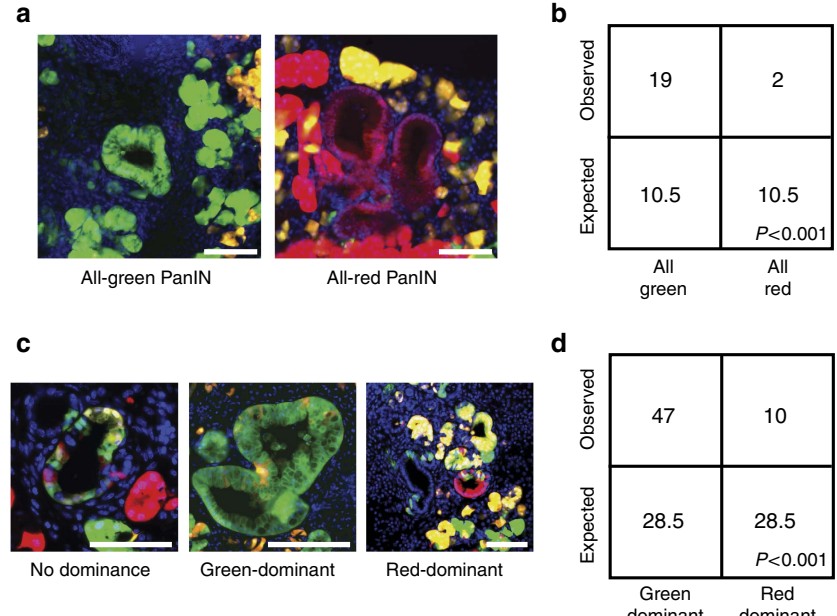

**Figure 7 | p53 loss promotes the initiation and expansion of low-grade PanINs.** (**a**) Low-grade uniform colour PanINs from 6-week-old *Pdx1-Cre-K-MADM-p53* mice showing *p53^{KO/KO}* (green, GFP +/tdTomato −), *p53^{WT/WT}* (red, GFP −/tdTomato +) and *p53^{KO/WT}* (yellow, GFP +/tdTomato +) cells. Representative images of all-green and all-red PanINs are shown. Blue, DAPI-stained nuclei. Scale bars, 100 μm. (**b**) Absolute quantification of observed all-green and all-red low-grade PanINs from 6-week-old *Pdx1-Cre-K-MADM-p53* mice ($n = 5$ mice total). Expected numbers are based on 1:1 stoichiometric ratio of green and red cell generation, and were not observed ($P < 0.001$, $\chi^2$-test). (**c**) Low-grade mixed-colour PanINs from 6-week-old *Pdx1-Cre-K-MADM-p53* mice showing green, red and yellow cells. Representative images of non-dominant, green-dominant and red-dominant low-grade PanINs are shown. Blue, DAPI-stained nuclei. Scale bars, 100 μm. (**d**) Absolute quantification of observed green-dominant and red-dominant low-grade PanINs from 6-week-old *Pdx1-Cre-K-MADM-p53* mice ($n = 5$ mice total). Expected numbers are based on 1:1 stoichiometric ratio of green and red cell generation, and were not observed ($P < 0.001$, $\chi^2$-test). A minority of low-grade PanINs (14 of 94) did not exhibit color dominance.

lineage-related cells during tumour progression. Using MADM, we observed green mesenchymal-like cells nearby but dispersed from green high-grade PanINs, some of which had lost expression of the epithelial marker cytokeratin 19 (Fig. 10a), in 6-week-old *Pdx1-Cre-K-MADM-p53* mice. Given the low frequency of high-grade PanINs or PDACs at this time point (~2.2 lesions per pancreas section per mouse), these invasive cells are likely to be derived from the PanIN epithelial cells. These findings are consistent with extratumoral invasion and epithelial-to-mesenchymal transition during putative preinvasive PanIN stages and corroborate previous work using *Pdx1-Cre; LSL-Kras^{G12D}/Kras^{WT}; p53^{flox/WT}* mice harbouring a *LSL-YFP* reporter[30]. Moreover, as MADM induces G2-X recombination sporadically at low efficiency, it offers a much more stringent evaluation of clonally related cells than traditional Cre/loxP-based reporters that may be prone to aberrant reporter expression from leaky transgenic Cre lines[31].

**Intratumoral cell dispersion in early tumours.** Aside from the extratumoral spread of cells from putative preinvasive tumours in the pancreas, we also observed remarkable intratumoral dispersion of lineage-related cells in early lung and pancreatic tumours. Rather than exclusively displaying adjacent clusters of expanding green and red cells as would be expected with the stationary growth of epithelial tumour cells, lung adenomas and low-grade PanIN lesions often displayed subclones in which cells were non-contiguous in tissue sections (Fig. 10b–d). We confirmed this cell dispersal in three dimensions by analysing intact tissues using multi-photon microscopy (Fig. 10e,f and Supplementary Movies 1–4).

To quantitatively assess cell dispersal, we calculated the mean distance between different green or red cells in tissue sections of lung adenomas from *K-MADM-p53* mice at 10-16 weeks p.i. Although labelled cells of the same colour were separated by an average of ~8.7 cell diameters (range 2.47–27.76, $n = 31$ tumours), *p53^{KO/KO}* cells surprisingly exhibited significantly decreased dispersal than *p53^{WT/WT}* cells (7.2 versus 10.2 cell diameters, $P = 0.013$, two-tailed paired Student's *t*-test). Indeed, as *p53^{KO/KO}* tumour cells progress, they appear to form more densely packed clones (Figs 2b,c and 3c). Together, these data support a model in which early tumours display subclonal dispersed growth, whereas tumour progression afforded by *p53* loss promotes more localized growth.

**Discussion**

In this study, we have used MADM to trace genetically distinct subclones within the same tumours to parse specific roles of p53 during different stages of lung and pancreatic tumorigenesis and to elucidate dynamic subclonal growth patterns in early epithelial tumours. MADM has several advantages over pre-existing mouse models of cancer. Unlike early models that induced simultaneous cooperating mutations in large numbers of cells[11,12,32], MADM permits sporadic and sequential mutagenesis of oncogenes and tumour suppressor genes, more faithfully mimicking the clonal genetic evolution observed in human cancers. Newer models have leveraged multiple recombination systems (Cre/loxP and FLP/FRT) to induce temporally separated mutagenesis events[33,34]. Mutagenesis events can be coupled to recombination-dependent fluorescent reporters, to permit tracing of genetically distinct clones[35–37]. By titrating down the dose and altering the timing of recombinase expression, sporadic genetic modulation can be induced. However, given that mutagenesis and fluorescent labelling occur through separate recombination events, decreased recombinase activity enhances the likelihood

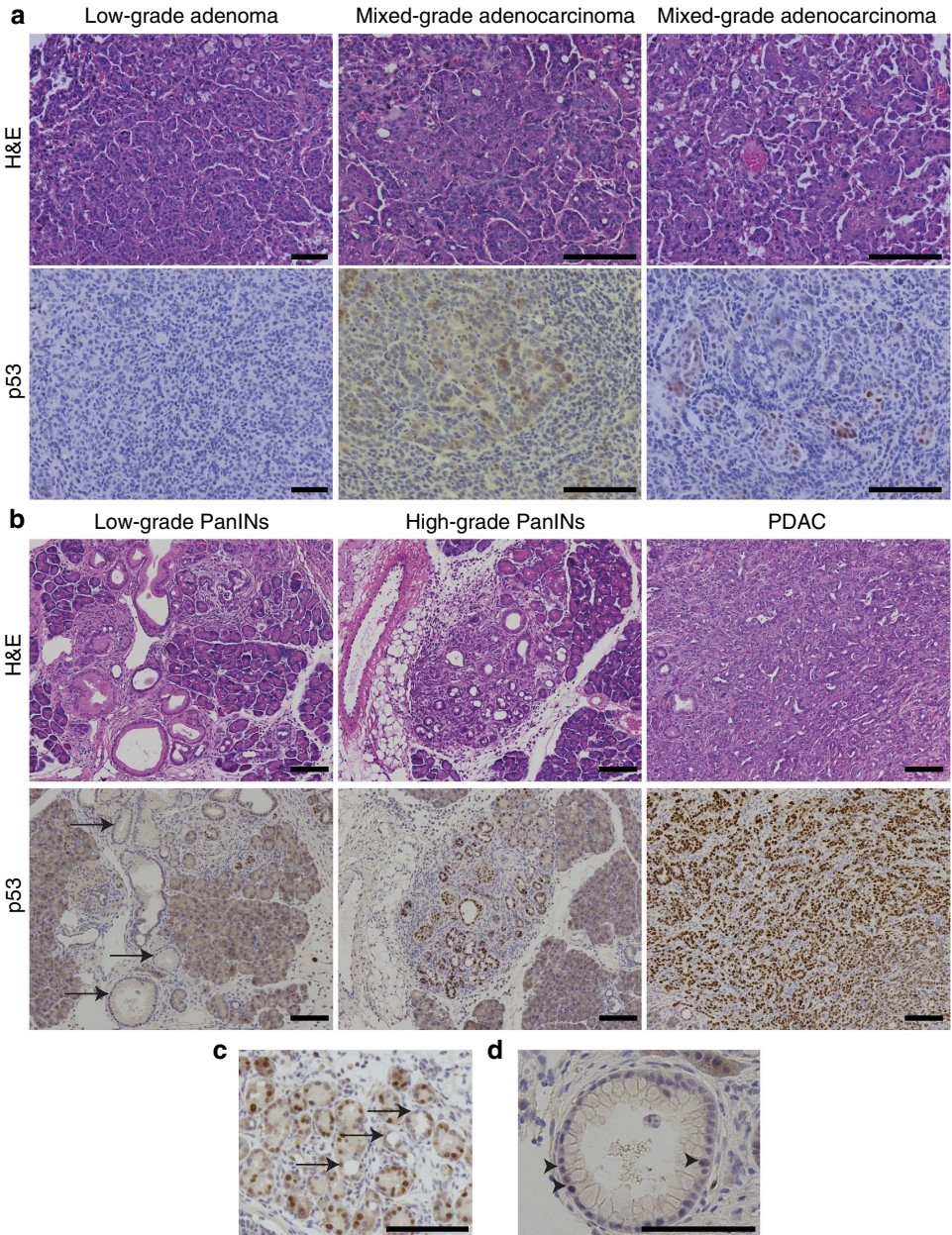

**Figure 8 | p53 expression in various stages of lung and pancreatic tumour progression.** (**a**) IHC for p53 in *LSL-Kras*$^{G12D}$*/Kras*$^{WT}$*; p53*$^{LSL-R172H/flox}$ low-grade adenomas and mixed-grade adenocarcinomas revealed p53 staining only in high-grade lung tumour cells. (**b**) IHC for p53 in *Pdx1-Cre; LSL-Kras*$^{G12D}$*/Kras*$^{WT}$*; p53*$^{LSL-R172H/WT}$ adult pancreas revealed increased p53 expression in higher-grade pancreatic lesions. Arrows show low-grade PanINs. (**c**) A subset of acinar-to-ductal metaplasia (ADM) cells expressed p53 (arrows). (**d**) A subset of low-grade PanIN cells expressed p53 (arrowheads). Scale bars, 100 μm (all).

of uncoupling of these events. In contrast, MADM-dependent labelling and mutagenesis occur through a single recombination event, maintaining the fidelity of the correlation of fluorescence protein expression and genotype. Moreover, MADM generates two genetically distinct subclonal population, permitting the tracing of lineage-related tumour suppressor gene wild-type and knockout clones within the same tumour.

By exploiting this feature of MADM, we have been able to define when p53 functions to suppress lung and pancreatic tumorigenesis. Genetic analyses on human cancers would suggest that p53 functions late during tumour progression, as *p53* mutations are principally identified in lung adenocarcinomas and high-grade PanINs or PDAC rather than their lower-grade precursors[17,18]. Our prospective evaluation of *p53* wild-type and

knockout clones suggest that this is indeed true during lung tumorigenesis and are consistent with experiments using *p53* reactivation models[14,15]. In contrast, p53 suppresses both the initiation and early expansion of pancreatic tumours. This conclusion would not have been drawn from existing genomic data from human tumours and validates the use of *in vivo* models to understand the molecular and cellular features that govern tumour progression.

The mechanisms that lead to differential induction of the p19ARF-p53 axis in early lung and pancreatic tumours remain unclear. In lung cancer, it is thought that enhanced oncogenic stress through *Kras* amplification and hyperactivation of the mitogen-activated protein kinase pathway may trigger p19ARF expression[14,15]. In contrast, *Kras* amplification is rarely

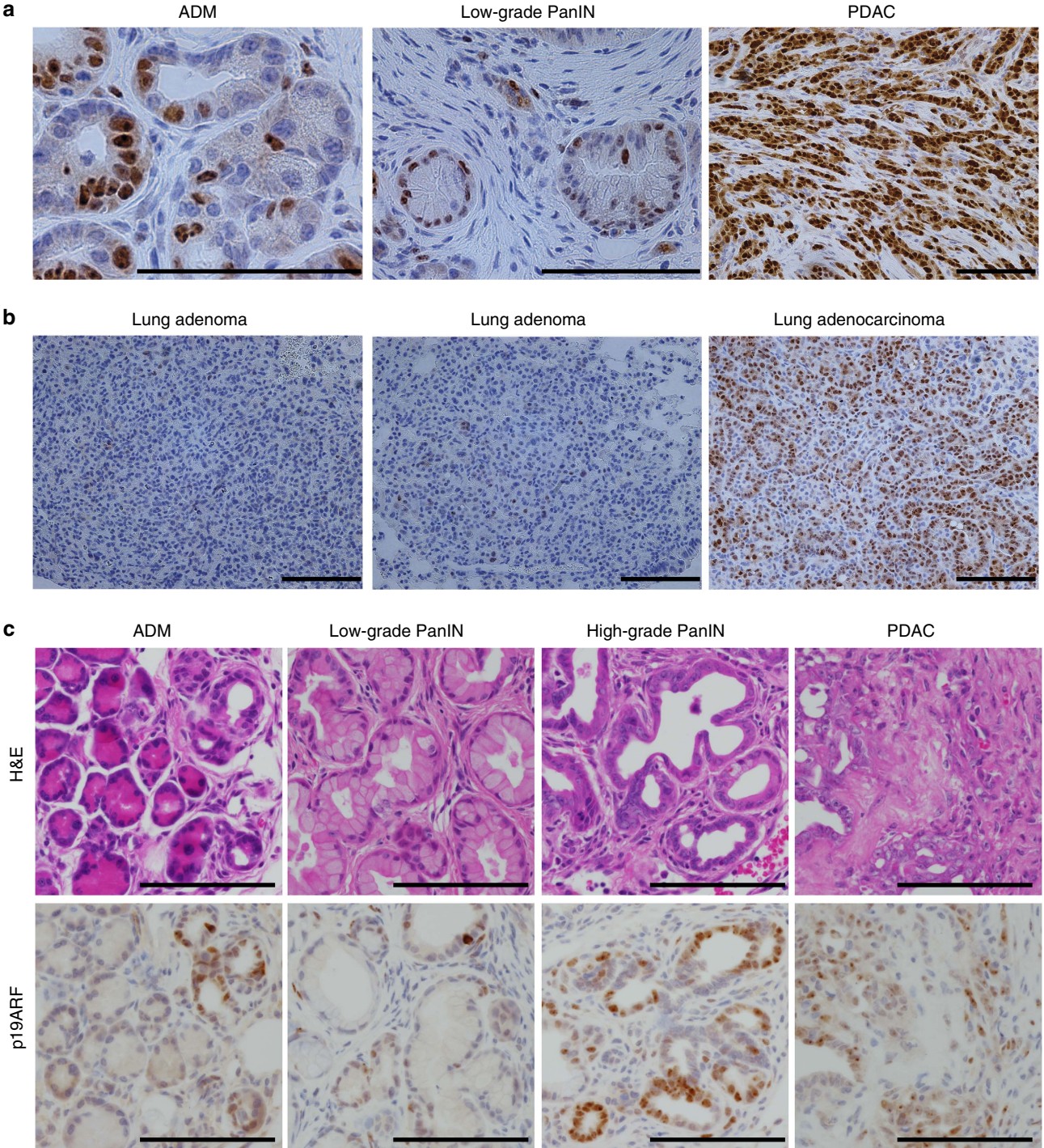

**Figure 9 | p19ARF is expressed in early lesions during pancreatic but not lung tumorigenesis.** (**a**) IHC for p19ARF in *Pdx1-Cre; LSL-Kras[G12D]/Kras[WT]; p53[LSL-R172H/WT]* tumours showed expression throughout pancreatic tumorigenesis from early acinar-to-ductal metaplasia (ADM) and PanIN lesions to PDAC. (**b**) IHC for p19ARF exhibited expression in *LSL-Kras[G12D]/Kras[WT]; p53[flox/flox]* lung adenocarcinomas but not adenomas. (**c**) IHC for p19ARF in *Pdx1-Cre; LSL-Kras[G12D]/Kras[WT]* (KC) tumours, lacking *p53* mutation, displayed expression throughout pancreatic tumorigenesis. These data suggest that *p53* mutation in **a** does not induce p19ARF by feedback upregulation, as has been previously described[28]. Scale bars, 100 μm (all).

observed in pancreatic cancer. Instead, activation of the phosphatidylinositol 3-kinase pathway downstream of oncogenic *Kras* may play a greater role in activating p19ARF during pancreatic tumorigenesis. Indeed, mutant *PIK3CA[H1047R]* expression in the pancreas phenocopies oncogenic *Kras* in terms of tumour progression and p19ARF expression[38]. In addition, Cre-mediated deletion of the PI3K effector Pdk1 in *Kras*-initiated pancreatic tumours reduces the induction of p19ARF[38].

Alternatively, tissue-specific mediators of ARF induction could also explain the differences in ARF expression between *Kras*-driven lung and pancreatic tumours. These include additional signalling pathways (for example, Notch[39]), epigenetic modifiers (for example, Bmi1 (ref. 28)) and transcription factors (for example, Dmp1 (ref. 40), AP-1 (ref. 41) and STAT3 (ref. 42)). Nonetheless, the capacity of *Pdx1-Cre; LSL-Kras[G12D]/Kras[WT]* mice to initiate tumours and develop high-grade pancreatic

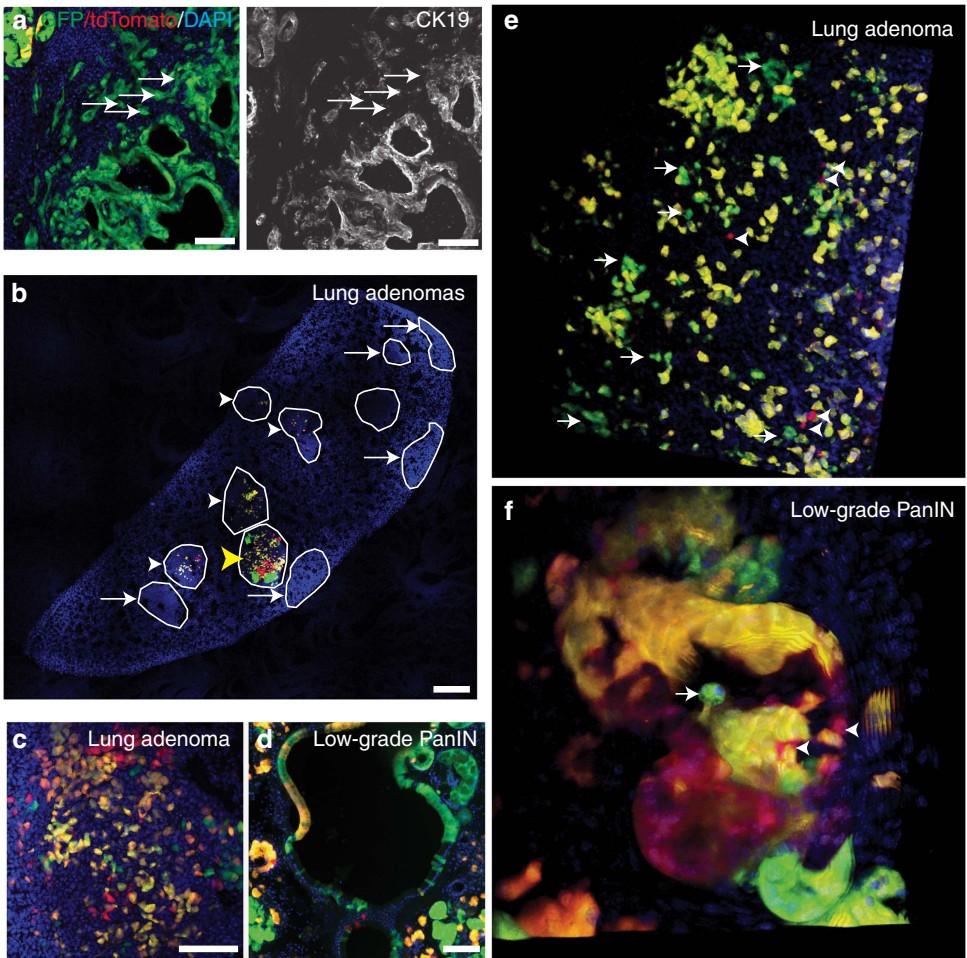

**Figure 10 | Extra- and intra-tumoral dispersal of early lung and pancreatic tumours.** (**a**) $p53^{KO/KO}$ (green, GFP+/tdTomato−) high-grade PanIN with extra-tumoral dispersal of CK19-negative cells (arrows) in a 6-week-old *Pdx1-Cre-K-MADM-p53* mouse. (**b**) Stitched confocal image of an entire lung lobe from a 10-week-p.i. *K-MADM-p53* mouse reveals multiple low-grade tumours, some harbouring fluorescently labelled cells (arrowheads) and others not (arrows). Most tumours showed dispersed labelling of green and red cells (white arrowheads), whereas some showed clusters of cells (yellow arrowhead). Scale bar, 500 μm. (**c**) Lung adenoma from a 16-week-p.i. *K-MADM-p53* mouse shows dispersed green, red and yellow cells. (**d**) Low-grade PanIN from a 6-week-old *Pdx1-Cre-K-MADM-p53* mouse shows dispersed green, red and yellow cells. (**e**) Three-dimensional (3D) rendering of multi-photon imaging of a lung adenoma from a 16-week-p.i. *K-MADM-p53* mouse showing non-contiguous green (arrows) and red (arrowheads) cells. (**f**) 3D rendering of multi-photon imaging of a low-grade PanIN from a 6-week-old *Pdx1-Cre-K-MADM-p53* mouse showing dispersed green (arrows) and red (arrowheads) cells. Blue, DAPI-stained nuclei. Scale bars, 100 μm (all, unless otherwise noted).

lesions despite wild-type *p53* and intact p19ARF expression (Fig. 9c) supports the existence of non-genetic mechanisms to evade p19ARF-p53-mediated tumour suppression. A better understanding of these mechanisms may offer novel therapeutic approaches to reinforce this tumour suppressive pathway to prevent tumour initiation and progression.

In addition to our *p53*-related findings, our results offer the first *in vivo* experimental evidence of spontaneous intratumoral dispersion of genetically distinct subclones during solid tumour progression. Genomic studies have revealed that most cells within a tumour harbour multiple truncal mutations with subclonal genetic heterogeneity occurring late during tumorigenesis[1,4–6]. How these truncal mutations propagate throughout a tumour remains poorly understood, although a recent computational model predicted intratumoral cell dispersion during tumour growth as a potential mechanism[43] and is consistent with our findings. The mechanism that leads to this dispersion phenomenon is unknown. Based on the overall low rates of proliferation in low-grade tumours (Supplementary Fig. 3), we hypothesize that labelled low-grade tumour cells divide at rates

comparable to neighbouring unlabelled cells permitting intermingling after several rounds of cell division of both cell types. In contrast, $p53^{KO/KO}$ cells that have progressed to higher grade divide faster than surrounding cells, allowing them to form more localized clusters. Alternative potential mechanisms for subclonal cell dispersion in early lesions include intratumoral cell migration, competition with neighbouring cells or immune cell clearance. The development of new approaches that permit long-term live *in vivo* imaging of these tumours could aid in exploring these explanations using these models.

## Methods

**Animal studies.** Animal studies were approved by the MIT Institutional Animal Care and Use Committee. All animals were maintained on a mixed background. *MADM11-GT* (Stock #013749), *MADM11-TG* (Stock #013751), *Pdx1-Cre* (Stock #014647) and *p53^{flox/flox}* (Stock #008462) mice were obtained from the Jackson Laboratory. *MADM11-TG,p53^{WT}/MADM11-TG,p53^{WT}* mice were crossed with *p53^{KO/WT}* mice, which carry a *p53*-null allele lacking exons 2–6 (ref. 44). Intercrossing the progeny *MADM11-TG,p53^{WT}/p53^{KO}* mice permitted recombination of the *p53*-null mutation onto the same chromosome as the MADM cassette. These *MADM11-TG,p53^{KO}/p53^{WT}* mice were subsequently

crossed to $MADM11\text{-}TG,p53^{WT}/MADM11\text{-}TG,p53^{WT}$ mice, to generate $MADM11\text{-}TG,p53^{WT}/MADM11\text{-}TG,p53^{KO}$. These mice were crossed with $LSL\text{-}Kras^{G12D}$ (ref. 32) mice to create $LSL\text{-}Kras^{G12D}/Kras^{WT}$; $MADM11\text{-}TG,p53^{WT}/MADM11\text{-}TG,p53^{KO}$ breeders.

$Pdx1\text{-}Cre$ or $Kras^{LA2},Rosa26\text{-}Cre^{ERT2}/Kras^{WT},Rosa26\text{-}Cre^{ERT2}$ mice[15,25] were crossed to $MADM11\text{-}GT/MADM11\text{-}GT$ mice to produce $Pdx1\text{-}Cre$; $MADM11\text{-}GT/+$ and $Kras^{LA2},Rosa26\text{-}Cre^{ERT2}/Kras^{WT}$; $MADM11\text{-}GT/+$ mice, which were thereafter intercrossed to generate $Pdx1\text{-}Cre$; $MADM11\text{-}GT/MADM11\text{-}GT$ and $Kras^{LA2},Rosa26\text{-}Cre^{ERT2}/Kras^{WT},Rosa26\text{-}Cre^{ERT2}$; $MADM11\text{-}GT/MADM11\text{-}GT$ as breeders. $LSL\text{-}Kras^{G12D}/Kras^{WT}$; $MADM11\text{-}GT,p53^{WT}/MADM11\text{-}TG\text{-}p53^{KO}$ ($K\text{-}MADM\text{-}p53$), $Pdx1\text{-}Cre$; $LSL\text{-}Kras^{G12D}/Kras^{WT}$; $MADM11\text{-}GT,p53^{WT}/MADM11\text{-}TG\text{-}p53^{KO}$ ($Pdx1\text{-}Cre\text{-}K\text{-}MADM\text{-}p53$) and control mice lacking $LSL\text{-}Kras^{G12D}$ ($Pdx1\text{-}Cre\text{-}MADM\text{-}p53$) or $p53$ mutation ($Pdx1\text{-}Cre\text{-}K\text{-}MADM$) were generated by crossing the above breeders. $Kras^{LA2},Rosa26\text{-}Cre^{ERT2}/Kras^{WT}$; $MADM11\text{-}GT,p53^{WT}/MADM11\text{-}TG\text{-}p53^{KO}$ ($Kras^{LA2},Rosa26\text{-}Cre^{ERT2}/Kras^{WT}$; $MADM\text{-}p53$) and $Kras^{LA2},Rosa26\text{-}Cre^{ERT2}/Kras^{WT}$; $MADM$ (lacking $p53$ mutation) mice were generated in parallel crosses. $Pdx1\text{-}Cre$; $LSL\text{-}Kras^{G12D}/Kras^{WT}$ (KC); $p53^{LSL\text{-}R172H/WT}$, KC; $p53^{flox/WT}$, KC; $p53^{LSL\text{-}R172H/flox}$ and KC; $p53^{flox/flox}$ pancreatic cancer mice and $LSL\text{-}Kras^{G12D}/Kras^{WT}$; $p53^{LSL\text{-}R172H/flox}$ and $LSL\text{-}Kras^{G12D}/Kras^{WT}$; $p53^{flox/flox}$ lung cancer mice were produced previously described[11,13,20]. Kaplan–Meier survival analyses were performed using Prism (GraphPad Software, Inc.). Mice were genotyped using tail DNA by the HotShot method. Genotyping primers and protocols are listed in Supplementary Tables 1 and 2, respectively.

**Lentivirus production and infection.** Lentivirus was produced by co-transfection of 293T cells with $pPGK\text{-}Cre$ lentiviral backbone and packaging vectors (delta8.2 and VSV-G) using TransIT LT-1 (Mirus Bio). Viral supernatant was collected at 48 and 72 h after transfection, filtered and concentrated by ultracentrifugation. Virus was resuspended in OptiMEM, titered and administered intratracheally to mice[24].

**Tamoxifen treatment of mice.** $Kras^{LA2},Rosa26\text{-}Cre^{ERT2}/Kras^{WT}$; $MADM\text{-}p53$ and $Kras^{LA2},Rosa26\text{-}Cre^{ERT2}/Kras^{WT}$; $MADM$ (lacking $p53$ mutation) mice at 5–10 weeks of age were injected intraperitoneally with tamoxifen (Sigma) at a dose of 9 mg per 40 g total body weight every other day for a total of three doses, to induce $Cre^{ERT2}$-mediated recombination.

**Tissue preparation and histology.** MADM mice were sacrificed by $CO_2$ asphyxiation and perfused with cold 4% paraformaldehyde (PFA; Electron Microscopy Sciences) in PBS. For proliferation studies, mice were intraperitoneally injected with EdU (Setareh Biotech) 1.5 h before sacrifice. Tissues were dissected, fixed overnight with cold 4% PFA, cryoprotected with 30% sucrose and embedded in OCT (Tissue-Tek). For whole-mount multi-photon imaging, fixed tissues were liberated from OCT by washing with PBS, incubated with 1 µg ml$^{-1}$ 4,6-diami-dino-2-phenylindole (DAPI; Life Technologies) for 1 h at room temperature (RT) or overnight at 4 °C and stored in cold PBS before imaging. For tissue section analyses, 5–30 µm sections were cut using a Leica cryostat, air dried for 30–60 min, washed three times with PBS, stained with DAPI (Life Technologies) for 5 min and mounted in Vectashield (Vector Labs) before imaging. For immunofluorescence staining, sections were blocked with 0.5% PNB (Perkin Elmer) and stained with primary antibody, donkey anti-rabbit Alexa 647 secondary antibody (Life Technologies A-31573, 1:500) and DAPI before mounting in Vectashield. Rabbit anti-CK19 (Abcam ab133496, 1:100) and rabbit anti-cleaved caspase-3 (Asp175) (Cell Signaling Technologies 9664, 1:100) primary antibodies were used. EdU was detected in tissue sections using a Click-iT EdU Alexa Fluor 647 Imaging Kit (Thermo Fisher Scientific).

CLARITY samples were prepared as previously described[26]. Briefly, MADM mice were perfused with a hydrogel monomer solution (4% acrylamide, 0.05% Bis, 0.25% VA-044 initiator, 4% PFA and 0.05% saponin in PBS) and intact tissues were fixed in the same solution for 72 h at 4 °C. Samples were degassed for 10 min under nitrogen, incubated at 37 °C for 4 h, rotated in clearing solution (200 mM boric acid and 4% SDS pH 8.5) and subject to electrophoretic tissue clearing in a custom-built chamber for 48 h at 55 °C and 10–15 V. Samples were incubated in 0.1% Triton X-100 in PBS for 3 days followed by an additional 3 days in FocusClear (CelExplorer Labs) before imaging.

IHC for p53 and p19ARF was performed as previously described[15]. Briefly, formalin-fixed paraffin-embedded sections were stained with haematoxylin and eosin or immunostained with rabbit anti-p53 (Novacastra NCL-p53-CM5p, 1:400) or rat anti-p19ARF (Santa Cruz Biotech sc-32748, 1:100) using Impress secondary antibody kits on a Thermo Scientific Autostainer 360. Endogenous wild-type p53 was not detectable in tissue sections using this antibody; thus, p53 IHC was performed on sections from mouse tumours harbouring a $p53^{R172H}$ mutation exhibiting p53 protein accumulation due to loss of negative feedback.

**Primary cell culture and immunocytochemistry.** Pancreatic tumour cells from $Pdx1\text{-}Cre\text{-}K\text{-}MADM\text{-}p53$ mice were dissociated using a protease cocktail containing collagenase (Worthington), dispase (Roche) and trypsin-EDTA (Invitrogen) in Hank's balanced salt solution, quenched with fetal bovine serum

(Hyclone) and passed through a 100 µm filter (Falcon) before plating in DMEM (CellGro) containing 10% fetal bovine serum and penicillin/streptomycin (VWR). Cells were grown in culture for 2–3 days and subject to FACS analysis using a Guava flow cytometry system (Millipore). Stable GFP + /tdTomato − (green) cell lines were obtained through serial passage and confirmed to be $p53^{KO/KO}$ by PCR genotyping of genomic DNA isolated using QuickExtract (Epicentre). For immunocytochemistry, primary cultured cells were grown on glass coverslips, fixed with PFA, permeabilized with 0.2% Triton X-100, blocked with 5% BSA in PBS and stained with rabbit anti-p53 primary antibody (Novacastra NCL-p53-CM5p, 1:200), donkey anti-rabbit Alexa 647 secondary antibody (Life Technologies A-31573, 1:500) and DAPI before mounting onto glass slides with Vectashield. Unlike with tissue sections, endogenous p53 was readily detectable by immunofluorescence on cultured cells.

**Imaging.** Live imaging of cultured cells was performed with a Nikon Eclipse TE2000-U light microscope and SPOT RT3 camera. Immunocytochemistry and immunofluorescence tissue section images were obtained using an Andor camera attached to a Nikon 80 Eclipse 80i fluorescence microscope using × 4, × 10 and × 20 objectives. Additional immunofluorescence tissue section images were obtained with an Olympus FV1200 Laser Scanning Confocal Microscope with × 10, × 20 and × 30 objectives. Whole-mount images of intact fixed tissues were taken using a Nikon Eclipse TE2000-U dissecting light microscope and SPOT RT3 camera. CLARITY samples were imaged at a depth of up to 0.5 mm using a Nikon A1R Ultra-Fast Spectral Scanning Confocal Microscope. Multi-photon imaging was performed on an Olympus FV1000MP inverted microscope with a × 25, numerical aperture 1.05 objective and a SpectraPhysics Maitai tai/sapphire laser at 840 nm. Collagen 1, collected as second harmonic-generated polarized light, DAPI, green fluorescent protein (GFP) and tdTomato images were collected in three photomultiplier tubes (PMTs) with band-pass filters of 425/30, 425/45 and 607/70. Optical sections 5 µm apart to a tissue depth of 100–200 µm were obtained. IHC sections were imaged using a Nikon Digital Sight DS-U3 camera. Image processing, including merging colour channels, three-dimensional reconstructions and movie generation, was performed with ImageJ (NIH). Figure 5a,b were published with permission from MIT.

**Quantification methods and statistical analyses.** All lung adenomas ($n = 132$ total) from a single section from each of five lungs derived from $K\text{-}MADM\text{-}p53$ mice infected at 10-16 weeks p.i. with lentivirus were classified for colour dominance as green-dominant ($n = 38$), red-dominant ($n = 31$) or no dominance ($n = 51$). An additional 12 tumours contained no labelled cells or only yellow cells without evidence of G2-X recombination. $\chi^2$-Test of green-dominant and red-dominant tumours was performed with the null hypothesis being an expected ratio of 1:1. All green and red cells from a random subset of tumours from 10 weeks p.i. ($n = 9$ tumours) and 16 weeks p.i. ($n = 8$ tumours) mice were quantified from alternating 30 µm sections through entire tumours (total tumour diameter ranged from 300 to 500 µm). Green-to-red cell ratios were calculated for each tumour and geometric means with 95% confidence intervals were plotted. Total number of single-labelled cells (sum of green and red cells) versus green-to-red cell ratio was plotted for all tumours and $R^2$ and $P$-values were calculated by linear regression.

For analyses of PanIN initiation and early expansion, all low-grade PanINs ($n = 92$) from a single section from each of five pancreata derived from 6-week-old $Pdx1\text{-}Cre\text{-}K\text{-}MADM\text{-}p53$ mice were classified for colour dominance as all green ($n = 19$), green-dominant ($n = 47$), no dominance ($n = 14$), red-dominant ($n = 10$) or all red ($n = 2$) (Fig. 7). $\chi^2$-Test of all-green and all-red or green-dominant and red-dominant low-grade PanINs was performed with the null hypothesis being an expected ratio of 1:1. Green-dominant and red-dominant high-grade PanINs and PDACs from these mice were analysed similarly.

For quantification of EdU incorporation, lung adenomas ($n = 28$) and lung adenocarcinomas ($n = 3$) were imaged using a × 10 objective and the number of DAPI + and EdU + nuclei were quantified using ImageJ for a single image per lung adenoma and a total of eight images from lung adenocarcinomas ($n = 3$ mice). The percentage of EdU + /DAPI + cells from each image was calculated, averaged for each grade of tumour and compared by two-tailed Student's $t$-test. For low-grade PanINs, × 10 objective images were obtained ($n = 2$ mice) and the number of total GFP + , EdU + /GFP + , total tdTomato + and EdU + /tdTomato + were manually counted. The percentage of EdU + /GFP + and EdU + /tdTomato + cells were calculated and compared by two-tailed Student's $t$-test.

Tumour spectrum analysis was performed on $Kras^{LA2}/Kras^{WT}$ mice by analysing histologic sections from the lung and pancreas after sacrifice for tumour-related morbidity. Some of these mice also harboured heterozygous or homozygous mutations in $Lep$ (leptin, $ob$) or $Lepr$ (leptin receptor, $db$), although these genotypes did not have an impact on the frequency of tumour types observed.

Cell-to-cell distance in lung adenomas was determined by calculating the mean distance between labelled nuclei based on Delauney triangulation using the Delauney Voronoi plugin for ImageJ. Two-dimensional mean distances were calculated separately for all green and red cells within tumours ($n = 31$) from a single tissue section from $K\text{-}MADM\text{-}p53$ mice infected at 10-16 weeks p.i. and normalized to the mean distance between adjacent cells on the same image. Green and red cell distances were compared by two-tailed paired Student's $t$-test. $P < 0.05$ was used as level of significance for all statistical analyses.

**Data availability.** The authors declare that the data that support the findings from this study are available within the article and its Supplementary Information files or available from the corresponding author upon request.

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

## Acknowledgements

We thank P.-Y. Chen for critical reading of the manuscript; A. Li and K. Mercer for technical assistance; K. Cormier and C. Condon from the Hope Babette Tang (1983) Histology Facility for IHC support; J. Wyckoff from the Swanson Biotechnology Center Microscopy Core facility for technical assistance with confocal and multi-photon imaging; N.R. Kerper for help with CLARITY experiments; and A. Berns, A. Lowy, L. Luo and H. Zong for mice. This work was supported by the Lustgarten Foundation, the Howard Hughes Medical Institute and, in part, by a Cancer Center Support (core) grant P30-CA14051 from the National Cancer Institute. M.D.M. is supported by a KL2/Catalyst Medical Research Investigator Training award (an appointed KL2 award) from Harvard Catalyst | The Harvard Clinical and Translational Science Center (National Center for Research Resources and the National Center for Advancing Translational Sciences, National Institutes of Health Award KL2 TR001100). The content is solely the responsibility of the authors and does not necessarily represent the official views of Harvard Catalyst, Harvard University and its affiliated academic healthcare centres, or the National Institutes of Health. T.T. is supported by a NIH Pathway to Independence Award (K99). V.G. is supported by a Jane Coffin Childs Memorial Fund Postdoctoral Fellowship. C.M.L. is supported by the Ludwig Center for Molecular Oncology Fund. T.J. is a Howard Hughes Medical Institute Investigator, the David H. Koch Professor of Biology and a Daniel K. Ludwig Scholar.

## Author contributions

M.D.M. and T.J. designed the study. M.D.M., K.J.D., K.M.C. and R.R. performed experiments. T.T. assisted with CLARITY experiments. V.G. and C.M.L. provided lung tumour sections for IHC. M.D.M. and T.J. wrote the manuscript with comments from all authors.
