## [Peer Review File · Nature Communications]

Reviewer #1 (Remarks to the Author): Expert in prostate cancer

The manuscript "Clonal dynamics following p53 loss of heterozygosity in Kras-driven cancers" uses a number of novel genetically engineered mouse models to study the effect of sporadic p53 LOH in lung and pancreas cancer. The study shows that p53 loss promotes advanced adenocarcinoma in both organs, while having a tumor-promoting role at early stages of carcinogenesis only in the pancreas. This study provides new understanding on the role of p53 loss in carcinogenesis, it highlights tissue-specific roles of this tumor suppressors and described a model of sequential mutagenesis that better approximates human tumorigenesis. Overall, only minor improvements are suggested, and the manuscript should be of broad interest to the scientific community.

Specific comments:

It would be interesting to know whether loss of p53 increases proliferation and/or protects from senescence (or apoptosis) in early lesions. This could easily be done by immunostaining.

Quantification of early stage lesions for loss of p53 should be shown in the main figures.

In Figure 3 and 4, it would be useful to label the panels so as to allow easier interpretation of the data.

Reviewer #2 (Remarks to the Author): Expert in p53

The authors have used MADM to study the role of p53 in tumor progression in mouse models of lung cancer and pancreatic cancer, and to monitor tumor cell dispersion in early stages of cancer progression.

Their main conclusion is that p53 plays different roles in the early stage of the two different cancer types: in pancreatic cancer it impedes hyperproliferation, whereas in lung cancer it does not affect hyperproliferation, and only comes into play later on, to prevent progression from early stage

(adenoma) to late stage (adenocarcinoma disease). In addition, they observed that lineage-related pre-cancerous cells were dispersed in the tissue, and were more dispersed when p53 was intact.

Overall, this is largely a descriptive study, which extends conclusions derived from less sophisticated models (simultaneous massive K-Ras activation + p53 ablation) to a more sophisticated setting (K-Ras activation followed sequentially by low-frequency p53 loss). As such, it provides rather little novel biological or mechanistic insights. Nevertheless, the work is very elegant and technically impressive, illustrates the power of the MADM methodology, and is likely to be noted by many cancer researchers. Therefore, publication in Nature Comm may still be considered.

Some technical comments:

1. Line 171. It should be validated that in normal ducts at early time points (before PanINs develop) there are equal ratios of all red and all green ducts.
2. The paragraph starting in line 181 is very confusing. Unless one reads carefully the legends of Supp. Fig. 7 and 8, it is not apparent anywhere from the text that these experiments are NOT done with the same mice as the rest of the study, but rather with mice containing a mutant p53 gene. What is being stained in these figures - wild type p53, or mutant p53? And what happens to p53 during progression in this model? How does this explain the observations in the previous figures, using a different mouse model?
3. Line 189. Interpretation of ARF results is not simple. ARF is a positive regulator of p53 protein stability, but ARF expression is negatively regulated by p53. Hence, p53 loss is expected to result in ARF upregulation. Combining this complication with the fact that the experiments were done in a model harboring a mutant p53 allele (see above), it is even harder to understand who is the chicken and who is the egg here. The authors need to explain and discuss the ARF results more clearly and more rigorously.
4. Line 216, 247. These interesting observations. However, there is no formal proof that the dispersed cells are indeed derived from the same single progenitor. There are presently genetic lineage marking methods that can get closer to this. Hence, this is at best only suggestive.

Reviewer #3 (Remarks to the Author): Expert in lung cancer

NCOMMS-16-07545-T

Mazumdar et al. "Clonal dynamics following p53 loss of heterozygosity in Kras-driven cancers "

The authors use technology of mosaic analysis with double markers (MDADM) to study the clonal dynamics in genetically engineered mouse models (GEMMs) of lung adenocarcinoma and pancreatic adenocarcinoma driven by mutant KRAS. The variable was the timing and effect of p53 knockout mutations which were tracked by green p53 KO/KO vs. red (p53 (wt/wt) cells. In both of these models previously they and others have shown a large number of adenomas developing which then in a few cases progress to full fledged adenocarcinomas. Their basic findings are that lung and pancreatic tumorigenesis differ, with p53 abnormalities not playing a role in the expansion of lung adenomas while p53 loss of function did play such a role in pancreatic adenoma expansion. In both cases, loss of p53 function was associated with development of truce carcinomas ("constrained progression to advanced adenocarcinoma...") In addition, they saw that histologically more advanced tumors had contiguous growth of subclones, while the early versions were "dispersed." They conclude their results show "cancer type-specific suppressive roles of p53 early in early tumor progression and offer novel insights into clonal growth patterns during tumor development."

Comments to the authors:

Overall this article is a technically tour de force and the results are interesting. However, the basic question is how much does it really teach us about similar tumor developments in humans and how the information could be used for early cancer detection, prevention, or subsequently in treatment? We have known for decades that mutations in KRAS and p53 can cooperate to give malignant behavior in both lung and pancreatic adenocarcinoma. Also it is not surprising that there could be differences in the biology of the two tumor types. From what is presented, I am not sure what the authors feel is how this information could help us deal with the human situation and what would be ways to test their findings in human tissues and patients to see if their mouse findings are relevant or of use in human tumor biology and clinical translation. A discussion of this by the authors would be instructive not only to see their thought processes, but such a discussion would immediately identify the great relevance of their findings to further studies in humans or, in contrast, show that the pathway is not clear. Clearly some studies in human materials would have been of interest. While such studies are probably impossible for pancreatic adenocarcinoma, they could have been attempted in the early pathogenesis of lung cancer. Given the role of cigarette smoking in both human lung and pancreatic cancer, it would have been interesting to know the effect of nicotine exposure (for example) on the behavior of the p53 clones given the interactive role of nicotine acting on nAChRs on p53 expression (for example if the GEMM models were exposed to nicotine would there be differences in cell dispersion and lineage effects?) The reason again is to relate this to the human situation. Also, given the recent impact of immunotherapy on these tumor types, did the immune system play a role in the dispersal or contiguous nature, or in differences between lung and pancreatic cancer? Finally, on a simple technical note, it would be nice to know the actual numbers of mice studied in each case. While there are details on the numbers of lesions, we have no idea about how many individual mice went into the data. In addition, I assume there were not gender differences but I could not find this clearly stated.

Reviewer #1 (Remarks to the Author): Expert in prostate cancer

The manuscript "Clonal dynamics following p53 loss of heterozygosity in Kras-driven cancers" uses a number of novel genetically engineered mouse models to study the effect of sporadic p53 LOH in lung and pancreas cancer. The study shows that p53 loss promotes advanced adenocarcinoma in both organs, while having a tumor-promoting role at early stages of carcinogenesis only in the pancreas. This study provides new understanding on the role of p53 loss in carcinogenesis, it highlights tissue-specific roles of this tumor suppressors and described a model of sequential mutagenesis that better approximates human tumorigenesis. Overall, only minor improvements are suggested, and the manuscript should be of broad interest to the scientific community.

We thank the reviewer for his/her comments.

Specific comments:

It would be interesting to know whether loss of p53 increases proliferation and/or protects from senescence (or apoptosis) in early lesions. This could easily be done by immunostaining.

As suggested by the reviewer, we performed immunostaining for proliferation (EdU) and apoptotic (cleaved-caspase 3 (CC3)) markers in early lung and pancreatic tumors. The percentage of EdU-positive $p53^{KO/KO}$ cells was significantly greater than EdU-positive $p53^{WT/WT}$ cells in low-grade PanINs (**Supplementary Fig. 3a**). The proliferation rate in low-grade adenomas was markedly lower than that observed in high-grade tumors (**Supplementary Fig. 3e**). Given the low proliferative rate of these low-grade tumors, it was rare to observe EdU+ $p53^{WT/WT}$ or $p53^{KO/KO}$ cells, and there were no notable differences between genotypes (**Supplementary Fig. 3c**). In contrast, lung adenocarcinomas, which were $p53^{KO/KO}$, showed significant numbers of EdU+ cells, while interspersed $p53^{WT/WT}$ and $p53^{KO/WT}$ cells were not EdU-positive (**Supplementary Fig. 3d**). We observed rare apoptotic cells in PanINs (**Supplementary Fig. 3b**) We did not observe apoptotic cells in lung adenomas (**Supplementary Fig. 3f**). Neither adenomas or PanINs showed clear differences in apoptosis for $p53^{WT/WT}$ and $p53^{KO/KO}$ cells (**Supplementary Fig. 3b**).

Quantification of early stage lesions for loss of p53 should be shown in the main figures.

We have converted multiple Supplementary Figures into main Figures to better align with the formatting requirements of *Nature Communications*. Consequently, we have converted quantification of early-stage lesions for loss of $p53$ in PDAC (previously **Supplementary Fig. 6**) into main **Fig. 7**.

In Figure 3 and 4, it would be useful to label the panels so as to allow easier interpretation of the data.

We have amended Figure 3 (now Figure 5) and 4 (now Figure 10) to include additional labels of the precise histologic stages and tumor types.

Reviewer #2 (Remarks to the Author): Expert in p53

The authors have used MADM to study the role of p53 in tumor progression in mouse models of lung cancer and pancreatic cancer, and to monitor tumor cell dispersion in early stages of cancer progression. Their main conclusion is that p53 plays different roles in the early stage of the two different cancer types: in pancreatic cancer it impedes hyperproliferation, whereas in lung cancer it does not affect hyperproliferation, and only comes into play later on, to prevent progression from early stage (adenoma) to late stage (adenocarcinoma disease). In addition, they observed that lineage-related pre-cancerous cells were dispersed in the tissue, and were more dispersed when p53 was intact. Overall, this is largely a descriptive study, which extends conclusions derived from less sophisticated models (simultaneous massive K-Ras activation + p53 ablation) to a more sophisticated setting (K-Ras activation followed sequentially by low-frequency p53 loss). As such, it provides rather little novel biological or mechanistic insights. Nevertheless, the work is very elegant and technically impressive, illustrates the power of the MADM methodology, and is likely to be noted by many cancer researchers. Therefore, publication in Nature Comm may still be considered.

We thank the reviewer for his/her comments.

Some technical comments:

1. Line 171. It should be validated that in normal ducts at early time points (before PanINs develop) there are equal ratios of all red and all green ducts.

As suggested by the reviewer, we have examined the frequency of green and red cells in normal ducts in our mice. We did not observe a difference in the ratios of all red and all green ducts in this context. We have added the following sentence to them main text to address this: “We first confirmed that there was no difference in the proportions of green and red normal duct cells, the putative cell-of-origin for PanINs.” While ductal cells are often thought of as the cell of origin for PanINs and PDAC, it is important to consider that murine studies have suggested that other cell types (acinar cells, islets cells, etc.) can give rise to pancreatic tumorigenesis in the right context and genetic system. While acinar cells were the most frequently labelled cell type by MADM (likely due to their overall greater abundance compared to other cell types in the pancreas), we did not observe a difference in the green and red cell frequencies of normal pancreatic cells of these histologies.

2. The paragraph starting in line 181 is very confusing. Unless one reads carefully the legends of Supp. Fig. 7 and 8, it is not apparent anywhere from the text that these experiments are NOT done with the same mice as the rest of the study, but rather with mice containing a mutant p53 gene. What is being stained in these figures - wild type p53, or mutant p53? And what happens to p53 during progression in this model? How does this explain the observations in the previous figures, using a different mouse model?

We thank the reviewer for his comments and appreciate the potential confusion that this may cause. Wild-type p53 is challenging to stain by IHC in tissue sections. To get around this, we used mice with *Kras*-initiated tumors that harbor a mutant allele of *p53* (*R172H*). Mutant *p53* is

stabilized in tumor cells, due in part to loss of negative feedback. We have characterized mutant *p53* expression (as a measure of endogenous *p53* expression) during various stages of tumor progression in lung and pancreatic tumorigenesis as shown in Figure 8 (previously **Supplementary Fig. 7**) These data demonstrate differences in *p53* expression in early lesions in these two cancer types. We have clarified this point in the main text with appropriate citations and the following language: “As wild-type *p53* is difficult to detect by immunohistochemistry on tissue sections with currently available antibodies, we took advantage of oncogenic *Kras*-driven lung and pancreatic cancer models harboring a *p53*^{R172H} mutant allele (*LSL-Kras*^{G12D}; *p53*^{R172H}), which demonstrate similar histologic progression to the MADM models. In these mice, mutant *p53* is stabilized, due in part to loss of feedback inhibition, and serves as a marker of endogenous *p53* expression.” We believe that similar *p53* expression patterns would be induced in the MADM models as well, as tumors are initiated by the same *Kras* mutant allele and share comparable histologic progression.

3. Line 189. Interpretation of ARF results is not simple. ARF is a positive regulator of p53 protein stability, but ARF expression is negatively regulated by p53. Hence, p53 loss is expected to result in ARF upregulation. Combining this complication with the fact that the experiments were done in a model harboring a mutant p53 allele (see above), it is even harder to understand who is the chicken and who is the egg here. The authors need to explain and discuss the ARF results more clearly and more rigorously.

We appreciate the complex feedback mechanisms involved in regulating ARF expression. To address the reviewer’s concern, we had stained for p19ARF in *Pdx1-Cre; LSL-Kras*^{G12D} (KC) mice (lacking *p53* mutation) and still observed ARF protein expression in acinar-to-ductal metaplasia (ADM) lesions and low-grade PanINs similar to what was observed in the *LSL-Kras*^{G12D}; *p53*^{R172H} mice. These data are shown in **Fig. 9c** (previously **Supplementary Fig. 8c**). While we discussed this in the figure legend, we have clarified this in the main text as follows: “As *p53* mutant cells may induce p19ARF by loss of negative feedback, we verified that p19ARF expression was observed in early pancreatic tumors even in the context of wild-type *p53* (**Fig. 9c**).”

4. Line 216, 247. These interesting observations. However, there is no formal proof that the dispersed cells are indeed derived from the same single progenitor. There are presently genetic lineage marking methods that can get closer to this. Hence, this is at best only suggestive.

We agree that we cannot definitively prove that the dispersed cells of the the same color truly derived from a single progenitor. Nonetheless, we would expect that since green and red cells are born from the same progenitor, they, and their progeny, should be adjacent to each other in early tumors. We do not observe this to be the case for the vast majority or red and green cells, consistent with evidence for intratumoral cell dispersal.

Reviewer #3 (Remarks to the Author): Expert in lung cancer

NCOMMS-16-07545-T

Muzumdar et al. "Clonal dynamics following p53 loss of heterozygosity in Kras-driven cancers "

The authors use technology of mosaic analysis with double markers (MDADM) to study the clonal dynamics in genetically engineered mouse models (GEMMs) of lung adenocarcinoma and pancreatic adenocarcinoma driven by mutant KRAS. The variable was the timing and effect of p53 knockout mutations which were tracked by green p53 KO/KO) vs. red (p53 (wt/wt) cells. In both of these models previously they and others have shown a large number of adenomas developing which then in a few cases progress to full fledged adenocarcinomas. Their basic findings are that lung and pancreatic tumorigenesis differ, with p53 abnormalities not playing a role in the expansion of lung adenomas while p53 loss of function did play such a role in pancreatic adenoma expansion. In both cases, loss of p53 function was associated with development of true carcinomas ("constrained progression to advanced adenocarcinoma...") In addition, they saw that histologically more advanced tumors had contiguous growth of subclones, while the early versions were "dispersed." They conclude their results show "cancer type-specific suppressive roles of p53 early in early tumor progression and offer novel insights into clonal growth patterns during tumor development."

Comments to the authors:

Overall this article is a technically tour de force and the results are interesting. However, the basic question is how much does it really teach us about similar tumor developments in humans and how the information could be used for early cancer detection, prevention, or subsequently in treatment? We have known for decades that mutations in KRAS and p53 can cooperate to give malignant behavior in both lung and pancreatic adenocarcinoma. Also it is not surprising that there could be differences in the biology of the two tumor types. From what is presented, I am not sure what the authors feel is how this information could help us deal with the human situation and what would be ways to test their findings in human tissues and patients to see if their mouse findings are relevant or of use in human tumor biology and clinical translation. A discussion of this by the authors would be instructive not only to see their thought processes, but such a discussion would immediately identify the great relevance of their findings to further studies in humans or, in contrast, show that the pathway is not clear.

We thank the reviewer for his/her insightful comments, recognizing that models are only useful if they are instructive for understanding the diseases, in this case human cancers, they were designed to model. We have included a more significant review of the relevance of our experiments and conclusions to human cancers in a new Discussion section.

Clearly some studies in human materials would have been of interest. While such studies are probably impossible for pancreatic adenocarcinoma, they could have been attempted in the early pathogenesis of lung cancer.

We appreciate the reviewer's desire for experiments using human samples, though it is unclear what precisely we would examine in these materials. Moreover, early tumors (lung adenomas and PanINs) are difficult to obtain, as most patients undergo diagnostic tissue biopsy or surgery with more advanced disease (adenocarcinoma). We believe the mouse models described in this study provide insight into lung and pancreatic tumorigenesis that cannot be easily identified in human materials. Histopathologic and sequencing studies have suggested that *p53* is mutated in adenocarcinomas but not low-grade adenoma/PanIN tumors. These analyses would suggest that *p53* principally functions as a tumor suppressor late in both lung and pancreatic tumorigenesis. In contrast, our data suggest that the ARF-*p53* axis also plays an important role in early pancreatic tumorigenesis, as early as tumor initiation. In addition, while computational modeling of genomic sequencing data has suggested that intratumoral subclonal cell dispersal may be an important mediator of tumor growth, this is difficult to prove in human samples, given the challenges in spatially tracking subclonal populations. Our MADM models permit analysis of lineage-related subclones and show that intratumoral cell dispersal occurs in early tumorigenesis. We have included these points and how the models may be instructive for understanding human cancers in the Discussion section.

Given the role of cigarette smoking in both human lung and pancreatic cancer, it would have been interesting to know the effect of nicotine exposure (for example) on the behavior of the *p53* clones given the interactive role of nicotine acting on nAChRs on *p53* expression (for example if the GEMM models were exposed to nicotine would there be differences in cell dispersion and lineage effects?) The reason again is to relate this to the human situation. Also, given the recent impact of immunotherapy on these tumor types, did the immune system play a role in the dispersal or contiguous nature, or in differences between lung and pancreatic cancer?

We appreciate that there are other modifying factors that may play a role in the phenotypes related to *p53* loss and subclonal dispersion observed in our model system. The reviewer raises interesting points regarding the potential for nicotine or the immune system to influence the biology we observe. Certainly, external factors, such as nicotine, could disrupt the p19ARF-*p53* axis in developing human tumors and could be a non-genetic mechanism of modulating tumor suppressive activities. In the present study, we used genetic tools to disrupt *p53* and study the relevant biology in both early and late lung and pancreatic tumorigenesis. Similarly, we have performed our experiments in fully immunocompetent mice, which we believe is an advantage in more faithfully modeling tumor biology over transplant models that could allow similar lineage tracing and analysis of clonal growth dynamics. Nonetheless, we believe that exploring these possibilities through nicotine administration of our mice or manipulating the immune system is beyond the scope of the present study.

Finally, on a simple technical note, it would be nice to know the actual numbers of mice studied in each case. While there are details on the numbers of lesions, we have no idea about how many individual mice went into the data. In addition, I assume there were not gender differences but I could not find this clearly stated.

The numbers of mice analyzed are stated in both the figure legends and methods sections. We did not observe gender differences when analyzing these mice.

Reviewer #1 (Remarks to the Author):

The revised form of this manuscript addresses all of my earlier concerns. This work should be of interest to the cancer community, and inform future mouse modeling of human disease.

Reviewer #2 (Remarks to the Author):

The authors have provided satisfactory explanations and have modified the text accordingly.

Acceptance is now recommended.